# Cell Penetrating Peptide-Based Self-Assembly for PD-L1 Targeted Tumor Regression

**DOI:** 10.3390/ijms222413314

**Published:** 2021-12-10

**Authors:** Feng Guo, Junfeng Ke, Zhengdong Fu, Wenzhao Han, Liping Wang

**Affiliations:** 1Key Laboratory for Molecular Enzymology and Engineering, Ministry of Education, Jilin University, Changchun 130012, China; fengguo19@mails.jlu.edu.cn (F.G.); kejf19@mails.jlu.edu.cn (J.K.); fuzd19@mails.jlu.edu.cn (Z.F.); hanwz18@mails.jlu.edu.cn (W.H.); 2School of Life Sciences, Jilin University, Changchun 130012, China; 3Engineering Laboratory for AIDS Vaccine, Jilin University, Changchun 130012, China

**Keywords:** peptide, self-assembly, gene therapy, PD-L1

## Abstract

Cell penetrating peptides (CPPs) are peptides that can directly adapt to cell membranes and then permeate into cells. CPPs are usually covalently linked to the surface of nanocarriers to endow their permeability to the whole system. However, hybrids with lipids or polymers make the metabolism much more sophisticated and even more difficult to determine. In this study, we present a continuous sequence of 18 amino acids (FFAARTMIWY(d-P)GAWYKRI). It forms nanospheres around 170 nm, which increase slightly after loading with siRNA and DOX. Notably, it can be internalized by cancer cells mainly through electronic interactions and PD-L1-mediated endocytosis. Compared with poly-l-lysine and polyethyleneimine, it has a much higher efficiency (about four times) of gene transduction while lowering toxicity. In the treatment of cancer, it causes apoptosis (21%) and inhibits the expression of SURVIVIN protein in vitro. In vivo, it shows good biocompatibility as there are no changes in mice’s body weight. When administering peptide-siRNA-DOX, tumor growth is inhibited the most (about three times). These results above prove the sequence to be a good candidate for gene therapy and drug delivery.

## 1. Introduction

In gene delivery systems, nonviral systems have gained wide attention because of the activation of oncogenes and excessive immunogenicity of viral vectors [1]. To absorb genes first and then penetrate the cell membrane, cationic polymers such as polyethyleneimine (PEI) [2,3,4,5] and poly-(l-lysine) (PLL) [6] have been proposed. However, significant toxicity can be observed in cells or animals after the administration of PEI. PEI was reported to activate Fasl-mediated antigen-induced cell death in the spleen [7]. Systemic administration of PEI caused liver necrosis and death, as well as the increase of small aggregates of both platelets and CD11-b-positive cells in the lung [8]. PLL also shows its very toxicity, which increases with the increment of its molecular weight [9]. Polyamidoamine (PAMAM) dendrimer, whose synthesis is complicated and costly [10], has potential cytotoxicity [11]. Thus, cationic nanocarriers with better biocompatibility such as peptide-based vehicles are wiser choices. Peptide-based self-assembly has been proposed to deliver cargos such as cyclic peptides [12,13], linear short peptides [14], and amphiphilic peptides [15]. Among them, di-phenylalanine (FF) is extracted from the Alzheimer’s β-amyloid polypeptides and can self-assemble in aqueous conditions. It is reported to have an easy production, functional versatility, biodegradability, biocompatibility, and non-immunogenicity [16,17,18]. FF has been decorated with other subunits from amino acids such as Cys [19], to polymers such as PCL [20]. It can boost the self-assembly of those systems.

More than 1700 CPPs have been reported, and the most common are TAT and PLL [21,22]. However, through only electronic interactions, off-targeting effects are downsides in types of cationic CPPs [23]. Thus, some targeted CPPs are proposed, such as octarginine (R8) targeting neuropilin-1 receptors [24]. RTMIWY(d-P) GAWYKRI is a continuous sequence of 14 amino acids designed by our lab to blockade the PD-1/PD-L1 process [25], which can be more facile to synthesize through solid-phase peptide synthesis technology. Meanwhile, its net charge is above 3 from pH 7.4 to 6.0, which means it is likely to absorb anionic nucleic acids and penetrate the cell membrane.

To increase the flexibility, we add two phenylalanines (FF) with two Alanines as the linker to boost its flexibility, to be 18 amino acids (FFAARTMIWY(d-P)GAWYKRI). Furthermore, we explore its capacity to load hydrophobic drug doxorubicin (Figure 1).

## 2. Results

### 2.1. Characterization of Peptide and Peptide-RNA/DOX

FF was reported to form nanotubes in the aqueous solution [26]. However, when connecting it to other peptides, the assembly may be influenced by other amino acids, such as Arg, Pro, etc. [15]. Thus, we first used SEM to observe if it can form nanostructures in an aqueous solution after sonication, and what types of structures they will be. Appendix A showed that the purity (97.43%) and molecular weight (2277.74) are qualified. As illustrated in Figure 2A,B, the peptide formed some nanospheres at the concentration of 0.5 mg/mL. Since there were Arg and Lys in the consequences, we estimated that it carried cationic ions, and Appendix A confirmed this. Meanwhile, hemolysis assays suggested its good biocompatibility in Appendix A.

Gene therapy is a method with great expectations to treat various diseases. This is due to its great potential to realize some affections that small-molecular inhibitors and antibodies will never possess, which were known as undruggable targets [27]. Knocking down gene expression, altering mRNA splicing, targeting trinucleotide repeat disorders, upregulating target genes, expressing genes, and editing the genome are examples of its peculiar functions [28]. Practically, some siRNA-based therapeutic products have been proved by the Food and Drug Administration (FDA), such as Patisiran, Givosiran, Lumasiran, and Inclisiran. However, the main obstacle of gene therapy is the billion-year-old lipid barrier [29], which prevents large, charged molecules such as RNA. Thus, many kinds of nano-systems are constructed to deliver RNA into cells.

Encouraged by the results above, we wonder if it could absorb anionic nucleic acids and package them into their cores. The gel retardation assays (Appendix A) showed that peptide (0.25 mg/mL) can completely package RNA (10 μM). In addition, it is stable in FBS after 24 h. Figure 2C,D showed the nanostructure of peptide-RNA, which was similar to the former. Furthermore, as the cores are hydrophobic [30], we wonder if this peptide could also package hydrophobic drugs. Thus, DOX was desalinated by triethylamine and then mixed with the peptide in DMSO. Finally, the solution was diluted by ddH_2_O to assist the formation of the peptide, followed by dialysis. As shown in Appendix A, almost all DOX can be loaded when the theoretical DLC was lower than 20%. However, when it came to a higher DLC (30%), only about 70% of DOX can be encased. Figure 2E,F showed there were no dramatic changes after loading with DOX. Therefore, all the following experiments were carried at 20% of DOX. Notably, the addition of RNA and DOX slightly changed the sizes of its nanostructure (Appendix A). Appendix A indicated that sonication boosted the assembly, and the decrease in fluorescence may be due to the π-π stacking, which suggested it to be a crucial force. However, some amino acids in the sequence such as A, I, and M provided hydrophobic force, and Y and W were also involved in π-π stacking. Additionally, R and K referred to electronic reactions [31].

### 2.2. Internalization of Peptide-RNA/DOX and Its Mechanism

To confirm the cell toxicity first, MTT assays were carried out. As shown in Appendix A, the concentration lower than 40 μM did not affect the viability of MDA-MB-231 cells, while the concentration lower than 20 μM did not affect HUVEC cells. Thus, the following assays were carried out at the concentration that was lower than 15 μM of peptide.

To explore whether peptides can carry RNA into cells, flow cytometry assays were carried out. As illustrated in Figure 3A,B, only RNA-FAM cannot enter cells while peptides can carry it together with being internalized. Thus, the fluorescence of RNA in the cell increased with time. Notably, the fluorescence diminished at 6 h, which may be due to the endogenous degradation in cells. Meanwhile, to further evaluate the toxicity, peptides were compared with commercialized cationic peptide-based (PLL 2kDa) and polymer-based (PEI) vehicles. MTT assays (Appendix A) indicated the high toxicity of PEI and similar non-toxicity of PLL (2 kDa–3 kDa). Though there were no significant differences between PLL and peptides at low concentrations, the toxicity of PLL increases with the molecular weight. Notably, Figure 3C,D illustrated the highest transduction efficiency of peptides among them after being incubated for only 30 min. To determine by what mechanism the peptide was internalized, inhibitors were used. Figure 3E,F showed β-cyclodextrin (disrupts the formation of the cholesterol domains) and heparin (combined with heparan sulfate proteoglycan) were the most important factors, which means the caveolae-mediated endocytosis and electrostatic interactions were the main factors separately. Additionally, the competition of PD-L1 protein significantly influenced the internalization, while wortmannin (blocks the formation of clathrin-coated vesicles) did not.

Programmed cell death-1 (PD-1)/PD-L1 ligand 1 (PD-L1) (CD279) is an inhibitory checkpoint, which can inhibit the activation of T and B cells in the tumor microenvironment [32]. Antibody drugs such as Nivolumab, Pembrolizumab, etc., are approved by the FDA. However, their drawbacks are obvious, such as high costs and poor tissue permeability. To determine whether it can still blockade PD-1/PD-L1 after assembly, exhausted T cell models were used. T cells were incubated with excessive tumor cells (incubated with IFN-γ to overexpress PD-L1) for 24 h to induce exhaustion, and the nanovesicles were added to blockade the process. Finally, the exhaustion degree was evaluated by the expression level of IL-2 after the stimulation of PMA/PHA. As shown in Appendix A, peptides can inhibit the process and increase the secretion of IL-2.

As the peptides were designed to co-load with RNA and DOX, CLSM and flow cytometry were used to observe the internalization of both cargos. Figure 4 and Appendix A showed the fluorescence of RNA-FAM (green) and DOX (red) in cells. The yellow indicated the co-localization of RNA and DOX. The intensity increased with time. Furthermore, PD-L1 competed with peptides, and thus the fluorescence decreased. Meanwhile, flow cytometry (Figure 5A,B) supported the results. Cells aggregated in the Q4 quadrant represented those for which neither RNA nor DOX was internalized. Q1 and Q3 quadrants represented those for which only DOX and RNA were internalized separately. Q2 quadrant represented those for which both RNA and DOX were internalized.

### 2.3. Anti-Tumor Effects In Vitro

Gene therapy is a method that has great expectations to treat various diseases. This is due to its great potential to realize some affections that small-molecular inhibitors and antibodies will never possess, which was known as undruggable targets [27]. Knocking down gene expression, altering mRNA splicing, targeting trinucleotide repeat disorders, upregulating target genes, expressing genes, and editing the genome are examples of its functions [28]. Practically, some siRNA-based therapeutic products have been proved by the Food and Drug Administration (FDA), such as Patisiran, Givosiran, Lumasiran, and Inclisiran. However, the main obstacle of gene therapy is the billion-year-old lipid barrier [29], which prevents large, charged molecules such as RNA. Thus, many types of nanosystems are constructed to deliver RNA into cells.

Since SURVIVIN is an inhibitor of apoptosis protein (IAP) that is overexpressed in nearly every type of cancer [33] and DOX is widely used in treating breast cancer, we used flow cytometry to detect the apoptosis of cells after administrations. Figure 6A,B showed peptide-RNA, peptide-DOX, and peptide-RNA-DOX caused cell apoptosis. The most apoptosis happened when RNA and DOX were both administrated. Real-time qPCR (Figure 6C) indicated the significant decrease in the expression of SURVIVIN, which was consistent with Western blot results (Figure 6D). These results suggested that peptide-RNA-DOX was a good system to treat breast cancer.

### 2.4. Anti-Tumor Effects In Vivo

After gaining the results above, we wondered if this medication effect would be consistent when curing tumors in vivo. Thus, we injected MDA-MB-231 cells into BALB/c nude mice and then we used our drug systems to treat them. As shown in Appendix A, two days after the injection of tumor cells, drugs were administrated once every two days. Figure 7A illustrated that there were no significant changes in body weight, which suggested the good biocompatibility of our nano vehicles. Figure 7B–D showed that both peptide-RNA and DOX can significantly restrain the growth of tumors. Notably, peptide-RNA-DOX can further inhibit tumor cells. Appendix A indicated that there were no obvious lesions on organs and the tumor mass in the lungs was diminished. However, some injuries can be observed on the heart when DOX was injected. These results supported the anti-tumor effects of our vehicles.

## 3. Discussion

Cationic peptides such as PLL and TAT were first used to absorb anionic nucleic acids, but their low transduction efficiency, low loading capacity, and high cell toxicity restrained their applications considerably. Recent research focused mainly on using them as one of the components in the delivery systems. For example, PLL for RNA absorption [34] and TAT for cell penetrating [35] are the most common applications. Qiu et al., has reported cell penetrating peptide (CPP3-CLP) for targeting A549 cells [34]. Zhang et al., conjugated TAT to chitosan [36]. However, these polysomes with unclear metabolism pathways still need to be explored. Therefore, tapping the potential of peptides themselves has appealed to great attention.

Peptides are pure amino acids without any additions such as chitosan, PEG, or DSPG, but can still target PD-L1. We hope to construct a system whose components are all from endogenous elements, which can be metabolized more easily. Though many pure peptide-based nano-systems have been proposed, very few of them can both absorb nuclei acids through electronic interactions and target a specific protein simultaneously. For instance, several cyclic peptide sequences can make the self-assembly by alternating α- and β-amino acids, β-amino acids, and δ-amino acids by molecular stacking and H-bonds between backbones [37]. However, they are quite rigid in structure, which makes it difficult to add ligands. Additionally, some artificial peptide-based viruses are cationic for both penetrating cell membranes and absorbing RNA [38]. However, this is only through electronic interactions, and they cannot target a specific ligand. Thus, we proposed a flexible system by connecting FFAA to a cationic targeting sequence. In this way, a pure peptide-based functional vehicle can be gained readily.

We suggest the capacity of FFAA to be a hydrophobic force even in an 18 continuous sequence. It preserves targeting ability after assembly. This formula of the combination of hydrophobic force (FFAA) and targeting sequence (with cationic amino acids) can be further explored since there are many peptide sequences screened by affinity. Furthermore, it is very facile to synthesize or can even be pre-designed in peptide display technology such as T4 bacteriophage, which is hard to realize when using non-amino acids components or cyclic peptide. Using this approach, many types of nanoparticles can be gained simultaneously, and the objects screened will change from small molecular peptides to nanostructures.

## 4. Materials and Methods

### 4.1. Reagent

Rink amide-AM resin, amino acids (with protection groups), 1-hydroxy benzotriazole (HOBt), benzotriazole-1-yl-oxytripyrrolidinophosphonium hexafluorophosphate (PyBOP), and N-methyl morpholine (NMM) were acquired from GL Biochem (Shanghai, China). RNA and RNA-FAM were purchased from Comate Bioscience (Changchun, China). Doxorubicin hydrochloride, agarose, MTT, wortmannin, heparin, β-Cyclodextrin, polyethyleneimine (PEI), Poly-(l-lysine) (PLL), Dimethyl sulfoxide (DMSO), and Triethylamine were obtained from Aladdin (Shanghai, China). GoldView II Nuclear Staining Dyes, SDS-PAGE gel preparation kit, DAPI, and TriQuick were gained from Solarbio (Beijing, China). Annexin V-FITC/PI apoptosis detection kits were purchased from BestBio (Shanghai, China). Anti-SURVIVIN antibody A5719 was obtained from Bimake (Houston, TX, USA). GAPDH (rabbit, AP0066) was gained from Bioworld (Minneapolis, MN, USA). Recombinant human PD-L1 (ab167713) was obtained from ABCAM (Shanghai, China). Dulbecco’s modified Eagle medium (DMEM) was purchased from Meilunbio (Dalian, China). Fetal bovine serum (FBS) was purchased from Kang Yuan Biology (Tianjin, China). GoScriptTM reverse transcription kits (A5001) and GoTaq qPCR Master Mix kit (A6001) were obtained from Promega (Madison, WI, USA). Annexin V-FITC/PI apoptosis detection kits were purchased from BestBio (Shanghai, China). RT Master Mix was gained from ABM (Vancouver, BC, Canada).

### 4.2. Synthesis of Peptide and Preparation of Nano Vehicles

The peptide, Arg-Thr-Met-Ile-Trp-Tyr-(d-Pro)-Gly-Ala-Trp-Tyr-Lys-Arg-Ile (peptide) was acquired from GL Biochem (Shanghai, China) and synthesized by sequential condensation of the amino acids from the C-terminus to the N-terminus through solid-phase synthesis. To synthesize nanospheres, peptides (0.5 mg) were dissolved into ddH_2_O (1 mL), followed by sonication for 10 min.

### 4.3. Gel Retardation Assay and Drug Loading Efficiency

To synthesize peptide-RNA, 10 μL peptides (0.25 mg/mL) were blended with 10 μL different concentrations of RNA, followed by sonication for 10 min, followed by incubation in FBS for 24 h. Then, the agarose electrophoresis was carried out to evaluate its RNA loading efficiency.

To synthesize peptide-RNA-DOX, peptides and RNA were dissolved in 20 μL DMSO to 2.5 mg/mL and 100 μM separately. DOX·Cl (10 mg/mL in DMSO) with the addition of triethylamine 25 μL was stirred for 2 h at 37 °C. Then, 10 μL DOX (different concentrations in DMSO) were added to the mixture of peptide and RNA. Afterwards, the concoction was added with 270 μL ddH2O, followed by sonication. Finally, the mixture was dialyzed in a dialysis bag (MW cutoff 3.0 kDa) for 24 h to remove unloaded DOX, triethylamine hydrochloride, and DMSO. Finally, the loading DOX concentration was calculated based on UV absorbance values at 480 nm with an ultraviolet spectrophotometer (UV2501, Shimadzu, Suzhou, China). Drug loading capacity (DLC) and loading efficiency (DLE) were calculated using the following formula:$$loading\;capacity\;\left(\%\right)=\frac{weight\;of\;loaded\;drug}{total\;weight\;of\;drug\;and\;vehicles}\times 100\%$$
$$loading\;efficiency\;\left(\%\right)=\frac{weight\;of\;loaded\;drug}{total\;weight\;of\;drug}\times 100\%$$

### 4.4. Characterization

Peptide, peptide-RNA, and peptide-RNA-DOX were characterized by scanning electron microscopy (SEM; Hitachi S-4700, Hitachi Ltd., Tokyo, Japan) and a zeta potentiometer (NANO ZS90, Malvern Panalytical, Malvern, UK).

### 4.5. Hemolysis Test

Blood from Sprague Dawley (SD) rats (2 mL) was treated with nano vehicles to different concentrations. The supernatant was gained by centrifugation (1500× *g*, 10 min) and measured at 545 nm.

### 4.6. Cell Culture and MTT Assays

MDA-MB-231, HCT116 cells were cultured in DMEM with 10% FBS at 37 °C with 5% CO_2_. For MTT assays, 1 × 10^4^ cells/well were seeded in 96 well plates. After 24 h, the drug was added and incubated for another 24 h, followed by the addition of MTT and detection at 450 nm.

### 4.7. Internalization of Nano Vehicles and Its Mechanism

MDA-MB-231 cells were incubated with peptide-RNA-FAM, peptide-RNA-DOX for different periods, the fluorescence was detected by flow cytometry and confocal laser scanning microscope (CLSM, ZEISS, Oberkochen, Germany) with or without the addition of inhibitors (wortmannin, heparin, β-cyclodextrin and PD-L1).

### 4.8. Cell Apoptosis

MDA-MB-231 cells were cultured (5 × 10^6^/well) with drugs for 24 h and were washed 3 times. Afterwards, cells were stained using an Annexin V-FITC kit after removal from the plate, followed by measurement of flow cytometry.

### 4.9. Quantitative Real-Time-PCR and Western Blot Assays

MDA-MB-231 cells were cultured (5 × 10^6^/well) with the administration of drugs. Afterwards, RNA was extracted with TriQuick and reversed by RT Master Mix, followed by adding a SYBR^®^ PCR kit. The proteins were extracted with a protein extraction kit. After the quantification of the concentration of proteins with the BCA kit, SDS-PAGE was carried. After the addition of primary antibodies and secondary antibodies, the PVDF membrane was visualized through the Tanon 5200 (Tanon Science & Technology Co., Ltd., Shanghai, China) imaging system.

### 4.10. In Vivo Administrations of Drug

Female BALB/c nude mice between 4 and 6 weeks old were gained from Weitong Lihua Experimental Animal Technology Co., Ltd. (Beijing, China). and raised in the Experimental Animal Center of Jilin University School of Life Sciences. A total of 5 × 10^6^ MDA-MB-231 cells were injected into the second fat pad on the right side of the mice. Two days later, drugs (DOX 1.5 mg/kg, RNA 0.5 mg/kg, once every two days) were administrated directly to the vicinity of the tumor. Each group consisted of 5 mice. After sacrificing, various physiological indexes were measured. Tumor volumes were calculated through the following equation: $V=L\times W^{2}/2$ (*L* for length, *W* for width).

### 4.11. Statistical Analyses

All in vitro data were obtained from independent experiments (three times at least) in which each of the treatment conditions were tested in triplicate. Tumor sizes were evaluated in independent experiments with *n* = 5 for each data point. All data are presented as mean ± standard deviations (SD). SPSS was used for statistical analysis. One-way-ANOVA was used to compare the differences between groups. *p* < 0.05 (*), *p* < 0.01 (**), *p* < 0.001 (***).

## 5. Conclusions

In this study, we report a continuous sequence of 18 amino acids (FFAARTMIWY(d-P) GAWYKRI). It formed nanospheres with the size of around 170 nm after sonication. When loaded with siRNA and DOX, it still formed nanovesicles and increased slightly in size. Measured by gel retardation assays and UV, we confirmed that 1mg peptide can load 6.6 μM siRNA and 0.2 mg DOX. In vitro experiments suggested its better biocompatibility and higher gene transduction efficiency (about four times) than both PLL and PEI. When using PD-L1 protein as a competitor, the transduction was hampered, which suggested that PD-L1 mediated endocytosis was important in this process. It can cause apoptosis (21%) in MDA-MB-231 cells when observed by flow cytometry. In vivo experiments also showed its good biocompatibility as there were no changes in mice’s body weight. Tumor growth was significantly inhibited with the administration of peptide-RNA-DOX (about three times). These results above suggested that this sequence can deliver siRNA and DOX for tumor regression.

## Figures and Tables

**Figure 1 ijms-22-13314-f001:**
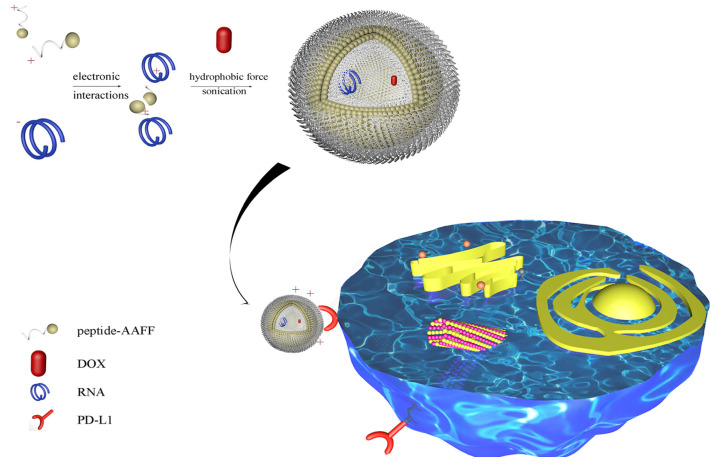
Schematic illustration of peptide-RNA-DOX for tumor regression. A PD-L1 targeted amino acids sequence was linked to Phe-Phe-Ala-Ala to form nanospheres. SiRNA and DOX were loaded through electronic interactions and hydrophobic force. This nano-system can target PD-L1 protein for tumor regression.

**Figure 2 ijms-22-13314-f002:**
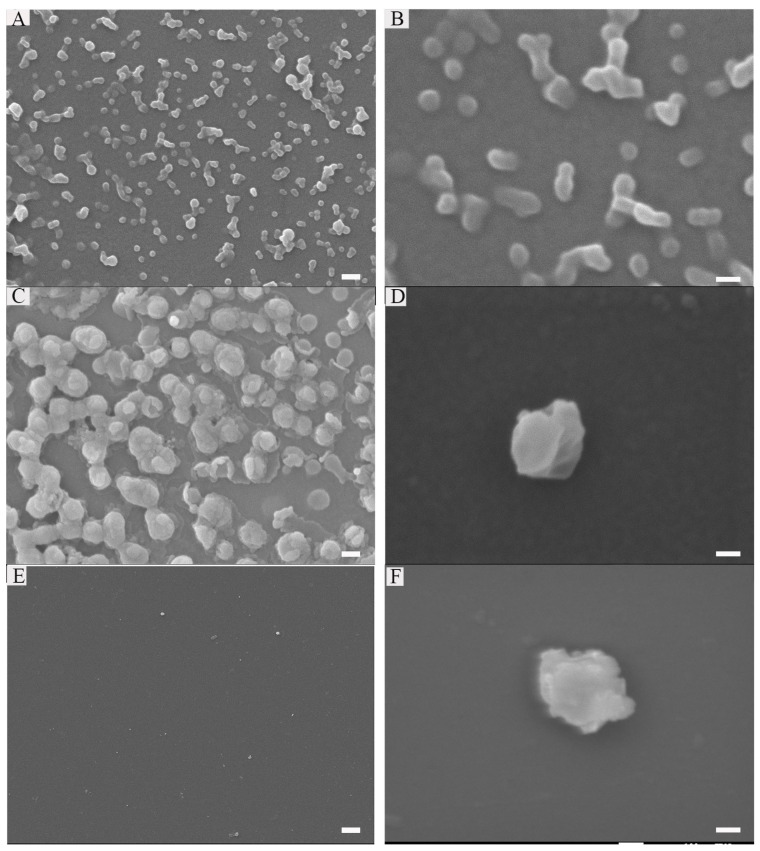
Characterization of peptide: 0.25 mg/mL peptide (**A**,**B**), Peptide (0.25 mg/mL)-RNA (5 μM) (**C**,**D**), peptide (0.25 mg/mL)-RNA (5 μM)-DOX (20 μg/mL) (**E**,**F**). Scale bars in the left array are 200 nm; scale bars in the right array are 100 nm.

**Figure 3 ijms-22-13314-f003:**
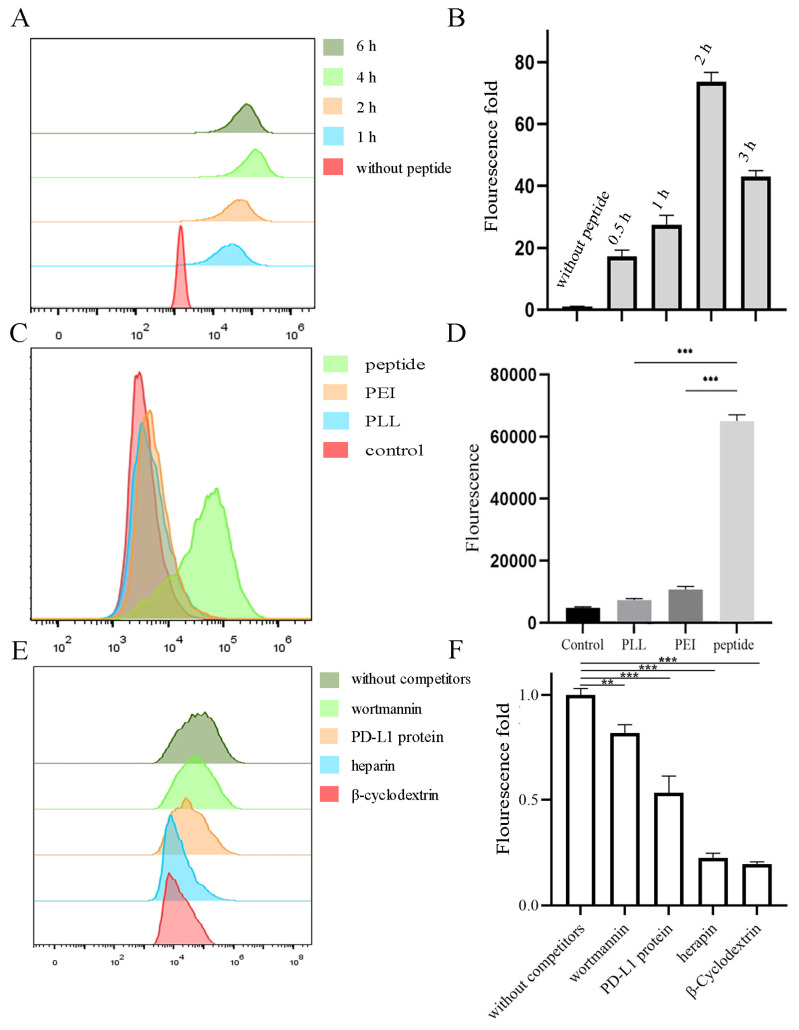
Cell internalization of peptide-RNA-FAM in MDA-MB-231 cells. (**A**,**B**) Peptide-RNA were incubated with cells, and the fluorescence was detected by flow cytometry after 1, 2, 4, and 6 h and quantified. (**C**,**D**) PLL, PEI, and peptide loaded with RNA. They were then incubated with cells for 0.5 h, followed by flow cytometry and quantified. (**E**,**F**) Cells were first incubated with inhibitors for 2 h followed by the addition of peptide-RNA. They were then detected and quantified. ** *p* < 0.01, *** *p* < 0.001, *n* = 3.

**Figure 4 ijms-22-13314-f004:**
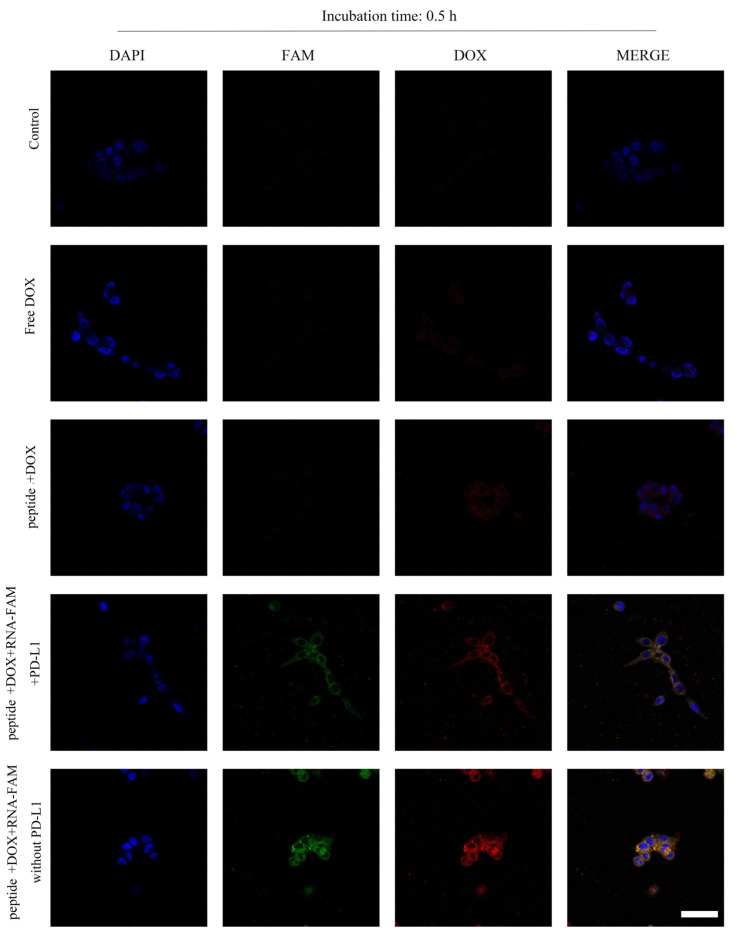
Cell internalization of peptide-RNA-DOX in MDA-MB-231 cells. CLSM images of cells after the incubation with PBS (the first line), free DOX (the second line), peptide-DOX (the third line), peptide-RNA-DOX with PD-L1 proteins (the fourth line), and peptide-RNA-DOX (the fifth line) for 0.5 h. Scale bar 50 μm.

**Figure 5 ijms-22-13314-f005:**
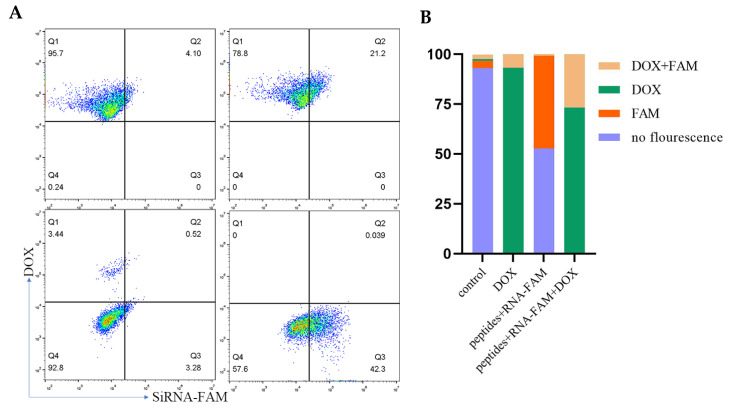
MDA-MB-231 cells were incubated with peptide-RNA-DOX for 0.5 h and then were detected by flow cytometry (**A**) and quantified (**B**).

**Figure 6 ijms-22-13314-f006:**
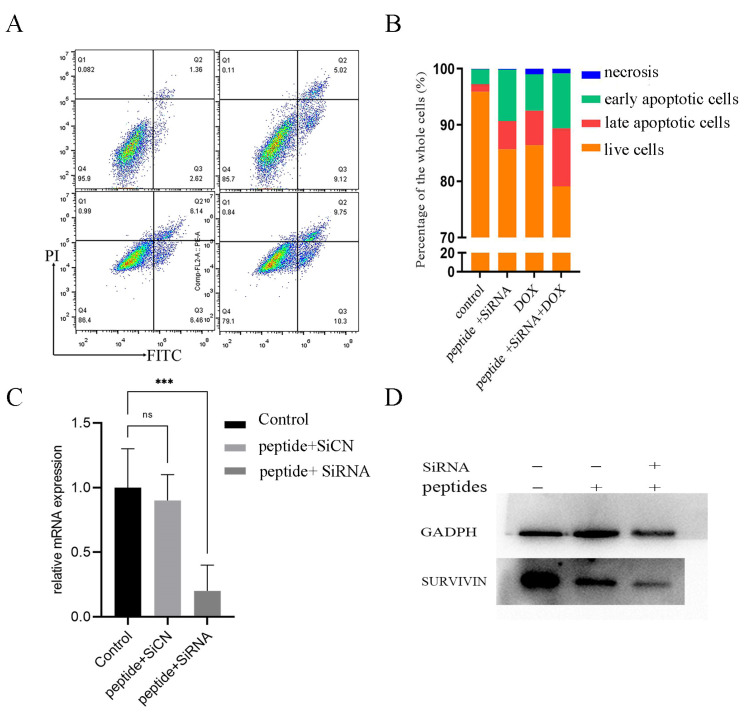
Cell apoptosis. (**A**) flow cytometry of cells after incubation with PBS (left top), peptide-RNA (5 μM) (right top), peptide-DOX (2 μg/mL) (left bottom), and peptide-RNA (5 μM)-DOX (2 μg/mL) (right bottom). Except for PBS, other groups caused apoptosis, and the last group was the most severe and the qualification (**B**). (**C**) q-rtPCR of cDNA reversed from mRNA extracted from cells after the administration of peptide-SiCN or peptide-SiRNA. (**D**) Western blot assays indicated that peptide did not decrease the expression of SURVIVIN while peptide-RNA did. *** *p* < 0.001, *n* = 3.

**Figure 7 ijms-22-13314-f007:**
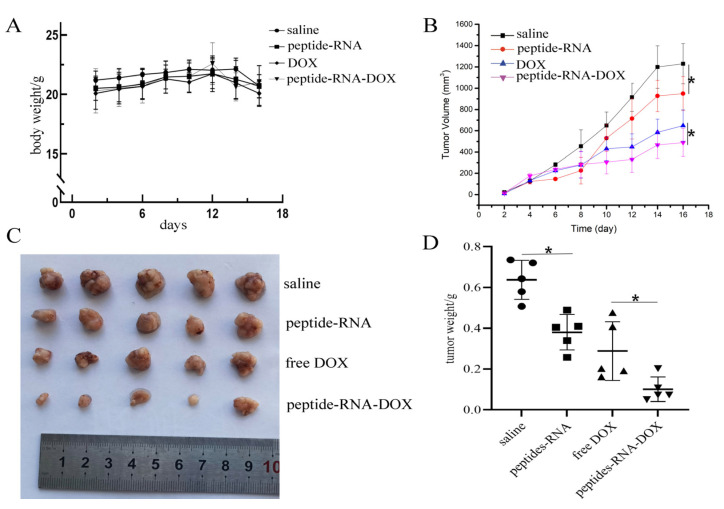
In vivo experiments. (**A**) Body weights of mice in four groups including PBS, peptide-RNA, free DOX, and peptide-RNA-DOX, which illustrated that there were no significant changes in different groups. (**B**) The tumor volumes of four groups during the therapy period. The chart illustrated that there were significant changes when using PBS and peptide-RNA. Additionally, peptide-RNA-DOX significantly decreased tumor volumes than DOX. (**C**) The photograph of tumors extracted from mice after tumors were fixed, which indicated the same trend as above. (**D**) Each weight of the fixed tumor. * *p* < 0.05, *n* = 5.

## Data Availability

Not applicable.

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
