# Peer review of "Cell Penetrating Peptide-Based Self-Assembly for PD-L1 Targeted Tumor Regression"

_ijms, 2021, doi:10.3390/ijms222413314_

Round 1

Reviewer 1 Report

Short peptides capable of crossing cell membranes are candidate molecules for the delivery of antitumor drugs, especially those directed to intracellular targets. The results are interesting and the authors used a number of approaches to validate their hypothesis. The methodology seems appropriate. However, the manuscript is confusing. In summary, the organization and flow of ideas highlight deficiencies that would need to be addressed. 

  1. The abstract must be improved. The main results and data of this study were not presented. Quantitative information must be included.
  2. The quality of the figures is not adequate.
  3. The peptide used in this investigation was previously described. Include information about it in the introduction section. What characteristics were decisive for selecting this peptide for this study?
  4. The order of presentation of the figures must be revised. Figure S5 is shown before S4. Additionally, some figures are not commented on in the text, for example, Figure S10 was not mentioned in the text. Please ensure all figures are described in the manuscript and are presented in the correct order.
  5. The results section includes some elements of the discussion. The current discussion presented is very poor. Considering that the manuscript has both sections, review the distribution and enrich the discussion section considering previous literature, comparisons, perspectives based on the findings of the article.
  6. Scheme 1 is a graphical abstract that summarizes the investigation. If this figure remains, it should be considered a figure. More information and context should be added to the legend. Likewise, the quality of the figure must be improved.
  7. It is not clear which cell type was used in the internalization assays. Please review the methodology and legend in figure 2.
  8. Authors should carefully review their conclusions. Which result it presents validates the following statement: "...However, electronic interactions influence the most in this process...."?
  9. The combinations were administered directly to the vicinity of the tumor.Why was this route of administration selected?
  10. All in vitro assays presented in this study were the the result of three independent (at least) experiments and all done in triplicate? If so, please this information. If not, please do more independent experiments.
  11. How many animals and how many repetitions were performed in the in vivo test?

Author Response

Point 1: The abstract must be improved. The main results and data of this study were not presented. Quantitative information must be included.

Response 1: Thanks a lot for your advice. We have rewritten the abstract as you suggested.

Point 2: The quality of the figures is not adequate.
Response 2: Thank you for pointing out the problem. All figures have been reinserted. Please see the figures in the text.

Point3: The peptide used in this investigation was previously described. Include information about it in the introduction section. What characteristics were decisive for selecting this peptide for this study?
Response 3: Thank you for your question. Actually, our lab has designed and synthesized a serry of 10 peptides as mentioned in the reference [1]. The peptide we chose in this paper has the decisive advantages as follows: 1) it has the highest affinity to PD-L1 protein; 2) its net charge is electropositive for absorbing electronegative RNA.

[1] Wang, K.; Song, Y.; Su, Y.; Liang, Y.; Wang, L. Effect of the hairpin structure of peptide inhibitors on the blockade of PD-1/PD-L1 axis. Biochem Biophys Res Commun 2020, 527, 453-457, doi:10.1016/j.bbrc.2020.04.018.

Point4: The order of presentation of the figures must be revised. Figure S5 is shown before S4. Additionally, some figures are not commented on in the text, for example, Figure S10 was not mentioned in the text. Please ensure all figures are described in the manuscript and are presented in the correct order.
Response 4: Thank you for pointing out the problem. We have carefully revised the order of figures and added texts about the unmentioned figures. Please see texts of each figure.

Point5: The results section includes some elements of the discussion. The current discussion presented is very poor. Considering that the manuscript has both sections, review the distribution and enrich the discussion section considering previous literature, comparisons, perspectives based on the findings of the article.
Response 5: Thank you for your advice. We have rewritten the discussion part to make the distribution of results and discussion appropriate. Meanwhile, we enrich the discussion as you suggest.

Point6: Scheme 1 is a graphical abstract that summarizes the investigation. If this figure remains, it should be considered a figure. More information and context should be added to the legend. Likewise, the quality of the figure must be improved.
Response 6: Thanks for your suggestion. We have changed scheme 1 to figure 1, and also added more information to the context.

Point7: It is not clear which cell type was used in the internalization assays. Please review the methodology and legend in figure 2.
Response 7: Thank you for your advice. We have added ‘in MDA-MB-231 cells’.

Point8: Authors should carefully review their conclusions. Which result it presents validates the following statement: "...However, electronic interactions influence the most in this process...."?
Response 8: Thank you for pointing out the problem. We have rewritten the conclusion part.

Point9: The combinations were administered directly to the vicinity of the tumor. Why was this route of administration selected?
Response 9: Thank you for your question. In situ injection of drug has been proposed by many papers.[2]-[4] Since the endocytic reticular system capture cationic substrates, we do not think this cationic peptide can escape from it. So, we chose to inject directly to tumor to gain better treatment efficiency.

[2]Korpan, N.N.; Xu, K.; Schwarzinger, P.; Watanabe, M.; Breitenecker, G.; Patrick, L.P. Cryo-Assisted Resection En Bloc, and Cryoablation In Situ, of Primary Breast Cancer Coupled With Intraoperative Ultrasound-Guided Tracer Injection: A Preliminary Clinical Study. Technol Cancer Res Treat 2018, 17, 1533034617746294, doi:10.1177/1533034617746294.

[3] Cai, L.; Du, X.; Zhang, C.; Yu, S.; Liu, L.; Zhao, J.; Zhao, Y.; Zhang, C.; Wu, J.; Wang, B.; et al. Robust immune response stimulated by in situ injection of CpG/alphaOX40/cGAMP in alphaPD-1-resistant malignancy. Cancer Immunol Immunother 2021, doi:10.1007/s00262-021-03095-z

[4] Lee, J.Y.; Kim, K.S.; Kang, Y.M.; Kim, E.S.; Hwang, S.J.; Lee, H.B.; Min, B.H.; Kim, J.H.; Kim, M.S. In vivo efficacy of paclitaxel-loaded injectable in situ-forming gel against subcutaneous tumor growth. Int J Pharm 2010, 392, 51-56, doi:10.1016/j.ijpharm.2010.03.033.

Point10: All in vitro assays presented in this study were the result of three independent (at least) experiments and all done in triplicate? If so, please show this information. If not, please do more independent experiments.
Response 10: Thank you for pointing out the problem. We have added ‘All in vitro data were obtained from independent experiments (three times at least) in which each of the treatment conditions were tested in triplicate. Tumor sizes were evaluated in independent experiments with n=5 for each data point. All data are presented as mean ± standard deviations (SD).’ in statistical analyses section.

Point11: How many animals and how many repetitions were performed in the in vivo test?
Response 11: Thank you for your question. Each group consists of 5 mice. We also add this information in section 4.10.

We would like to thank the referee again for taking the time to review our manuscript.

Reviewer 2 Report

Dear Authors,

 The manuscript “Cell penetrating peptide-based self-assembly for PD-L1 targeted tumor regression” contains numerous biological methods. The subject of work is interesting and important today. The methodology of research CPPs and their self-assemble into nanospheres and target PD-L1 proteins are provided on good level. However, the article require minor corrections before the publishing, according with the comments below:

  1. The article describes the research of only one peptide. The term 'peptides' is used in many places in the work at the plural form, rather it should be in the singular: 'peptide'.
  2. Please do not use short forms e.g. ‘What’s’ (in abstract) it is inappropriate in the formal style of writing scientific texts.
  3. The quality and resolution of some figures have to be higher. The most important point are listed below:
  • On scheme 1 the word above/under the arrow are illegible, please make large font.
  • The legend on Figure 2 especial part A, C, E have to be write larger font.
  • Figure 3: The separation of part A and B as independent figure would make them better readable.
  • Figure 4C is hard to read, it should be large.
  • Figure 5 A and B – please enlarge the legend.
  • Figure S3 – please enlarge the legend.
  1. Please check all text carefully, ‘spaces’ are missing in many places. Especially before the citation bracket.
  2. The description of peptide synthesis is very poor. Please add more information about: origin of each amino acid (name of company, state and country, level of piurity). Plugs blocking the appropriate groups and how to remove them.
  3. Please also add a description of the program that was used to determine the net charge of the peptide. How was obtain the Figure S3?
  4. The summary and discussion should be extend and include a comparison of the results obtained with other, for example, previously tested peptides.
  5. The supporting information file should contain the first title page, with name of authors and so on.
  6. Please add the abbreviation list.

Author Response

Point 1: The article describes the research of only one peptide. The term 'peptides' is used in many places in the work at the plural form, rather it should be in the singular: 'peptide'.

Response 1: Thank you a lot for your advice. We have changed the term 'peptides' to peptide both in text and figures. Also, the text and figures in supplement have been revised.

Point 2: Please do not use short forms e.g. ‘What’s’ (in abstract) it is inappropriate in the formal style of writing scientific texts.Response 2: Thank you for pointing out the problem. We have corrected all the short forms in the manuscript.

Point 3: The quality and resolution of some figures have to be higher. The most important points are listed below:

  • On scheme 1 the word above/under the arrow are illegible, please make large font.
  • The legend on Figure 2 especial part A, C, E have to be write larger font.
  • Figure 3: The separation of part A and B as independent figure would make them better readable.
  • Figure 4C is hard to read, it should be large.
  • Figure 5 A and B – please enlarge the legend.
  • Figure S3 – please enlarge the legend.

Response 3: Thank you very much for your advice. The changes are listed as below:

  • Characters in scheme 1(now figure 1) have been enlarged.
  • The legend of figure 2 (now figure 3) has been adjusted.
  • Figure 3 is been separated to figure 4 and 5 now. Text involving are corrected.
  • Figure 4C (now figure 6C) has been adjusted.
  • The legend in Figure 5A and B (now figure 7A and B) has been enlarged.
  • The legend in Figure S3 (now figure S6) has been adjusted.

Point 4: Please check all text carefully, ‘spaces’ are missing in many places. Especially before the citation bracket.
Response 4: Thank you for pointing out the problem. We have added the missing ‘spaces’ in the right places.

Point 5: The description of peptide synthesis is very poor. Please add more information about: origin of each amino acid (name of company, state and country, level of piurity). Plugs blocking the appropriate groups and how to remove them.
Response 5: Thank you for your advice. The peptide was purchased from GL Biochem (Shanghai, China). And we have added this information to section 4.2. The product we gained was evaluated by HPLC-MS, which showed that the purity was 97.43% and molecular weight was 2277.74. (Figure S1)

Point 6: Please also add a description of the program that was used to determine the net charge of the peptide. How was obtain the Figure S3?
Response 6: Thank you for your question. The net charge was calculated by the online tool called ‘Concentration polypeptide property calculator’ from the company NovoPro. We have added this information in the context. Figure S3 was obtained through Malvern NANO ZS90. We also add this information in the text.

Point 7: The summary and discussion should be extended and include a comparison of the results obtained with other, for example, previously tested peptides.
Response 7: Thank you for your suggestion. We have rewritten the discussion part. Please see it in the manuscript.

Point 8: The supporting information file should contain the first title page, with name of authors and so on.
Response 8: Thank you for your advice. We have added relating information to the first page of the supplement.

Point 9: Please add the abbreviation list.
Response 9: Thank you for your suggestion. We have added the abbreviation list after Conclusion.

We would like to thank the referee again for taking the time to review our manuscript.

Round 2

Reviewer 1 Report

The authors satisfactorily answered most of the concerns. I recommend the publication of this manuscript, however I insist that the quality of the figures still needs to be improved. The bar scale of the microscopy figures are not visible, the font size of the figures is not standardized. Some figures look blurry. 

Author Response

Response to Reviewer 1 Comments

Point 1:  I recommend the publication of this manuscript, however, I insist that the quality of the figures still needs to be improved. The bar scale of the microscopy figures is not visible, the font size of the figures is not standardized. Some figures look blurry. 

Response 1: Thank you very much for your suggestion. We have made the following adjustment: (1) all figures have been replaced with a higher resolution. (2) Scale bars in Figures 4, S11, and S13 have been enlarged. (3) the font size of figures 3, 5, 6, and 7 have been adjusted.

Thank you very much for your advice.